# Improved video VAE for Latent Video Diffusion Model

## Abstract

Variational Autoencoder (VAE) aims to compress pixel data into low-dimensional latent space, playing an important role in OpenAI's Sora and other latent video diffusion generation models. While most of existing video VAEs inflate a pretrained image VAE into the 3D causal structure for temporal-spatial compression, this paper presents two astonishing findings: (1) The initialization from a well-trained image VAE with the same latent dimensions suppresses the improvement of subsequent temporal compression capabilities. (2) The adoption of causal reasoning leads to unequal information interactions and unbalanced performance between frames. To alleviate these problems, we propose a keyframe-based temporal compression (KTC) architecture and a group causal convolution (GCConv) module to further improve video VAE (IV-VAE). Specifically, the KTC architecture divides the latent space into two branches, in which one half completely inherits the compression prior of keyframes from lower-dimension image VAEs while the other half involves temporal compression into the 3D group causal convolution, reducing temporal-spatial conflicts and accelerating the convergence speed of video VAE. The GCConv in above 3D half uses standard convolution within each frame group to ensure inter-frame equivalence, and employs causal logical padding between groups to maintain flexibility in processing variable frame video. Extensive experiments on five benchmarks demonstrate the SOTA video reconstruction and generation capabilities of the proposed IV-VAE. The source code and weights will be made available to the public.

## 1 INTRODUCTION

In recent years, video generation has made significant advances in both academia and industry, showcasing cinematic-level visuals and the potential of world simulator (*e.g.*, OpenAI's Sora (OpenAI, 2024)). In particular, diffusion optimization methods on latent space have garnered widespread attention and dominated the video generation field due to their high efficiency and stability, resulting in a series of outstanding Latent Video Diffusion Models (LVDMs), *e.g.*, Stable Video Diffusion (SVD) (Blattmann et al., 2023), Open-Sora (hpcaitech, 2024), Open-Sora-Plan (Chen et al., 2024).

To achieve a bidirectional mapping between the high-dimensional pixel space and the low-dimensional latent space, Variational Autoencoder (VAE) (Kingma, 2013) are employed to encode the original video continuously into the latent space, along with spatial and even temporal compression. The compression rate and reconstruction quality of VAE directly determines the information effectiveness of the video corpus, which plays an important role in the exploration of LVDMs.

As the most commonly used video VAE, SVD-VAE (Blattmann et al., 2023) freezes the weights of the 2D encoder based on the image VAE (Rombach et al., 2022) and retrains a temporal decoder containing 3D convolutional blocks, which enables the feature interaction across different frames and significantly reduces flicker artifacts. Unfortunately, this method struggles to effectively eliminate redundancy in the temporal dimension, limiting the LVDMs to perceive and generate long sequences of video content.

To address the aforementioned issues, several works (Yang et al., 2024; hpcaitech, 2024; Chen et al., 2024; Qin et al., 2024; Zhao et al., 2024) have proposed to construct a 3D VAE with both spatial and temporal compression capabilities, which can be summarized in the following three points. In terms of model structure, the UNet network (Ronneberger et al., 2015) with strong pixel-perception

capabilities is still adopted, consistent with the classic image VAE series (*i.e.*, Stable Diffusion). For the operator implementation, most of the methods (Yang et al., 2024; hpcaitech, 2024; Chen et al., 2024; Qin et al., 2024) adopt 3D causal convolution with only forward sequence correlation, which can encode the image information independently (Yu et al., 2023b) while ensuring the continuity of the long sequence video encoding. During the optimization process, a well-trained 2D image VAE, focusing only on spatial compression, is used as an initialization for 3D video VAEs with equivalent latent channel dimensions (hpcaitech, 2024; Chen et al., 2024; Qin et al., 2024; Zhao et al., 2024), which can achieve better results compared to training a temporal-spatial compressed video VAE from scratch (Yu et al., 2023a). Although existing methods achieve video reconstruction under 4x time compression, we find that there remain deficiencies in the above three key processes, limiting the improvement of the video quality and continuity in reconstruction.

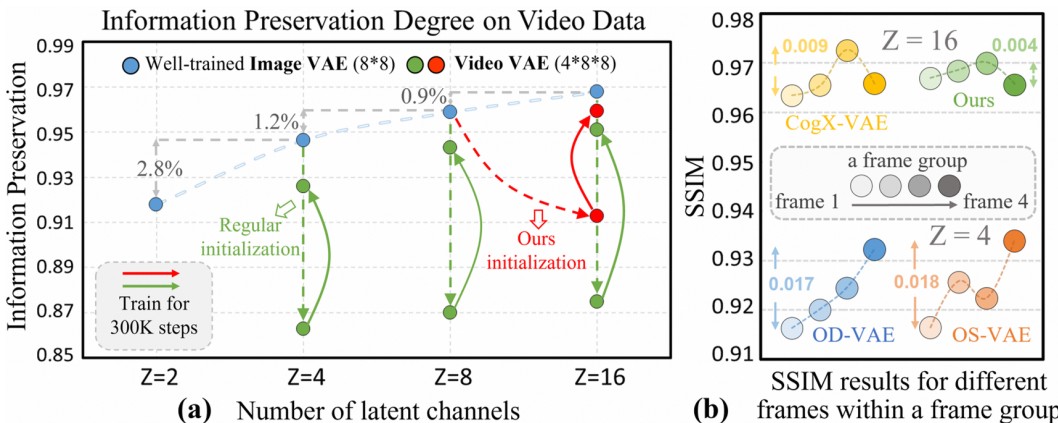

Figure 1: (a) **Information preservation degree** of reconstruction results for different VAEs on video data. (b) **Performance of different frames within a frame group** on kinetics-600 dataset. See Appendix A.1 and A.2 for the specific calculation process and definitions of the two diagrams.

First, initialization from a well-trained image VAE with the same latent dimension cannot support further 4x time compression, leading to a sharp decline in initial spatial compression performance and slow improvement in subsequent temporal compression. As shown in Fig. 1(a), we calculate the reconstruction similarity of different VAE as the information preservation degree on video data (See Appendix A.1). We note that: (1) As the latent channel number $Z$ increases, the spatial compression gain produced by the image VAE gradually diminishes (blue dashed lines), and even the spatial compression capability provided by an image VAE with dimension $Z/2$ is sufficient for a temporal-spatial compressed video VAE with dimension $Z$. (2) The video VAE that inherits image weights of a lower dimension $Z/2$ and uses the remaining latent channels to store the spatial information of an extra frame, i.e., acquiring the initial 2× time compression capacity (red curves), converges even better and faster than one that inherits weights of the same dimension $Z$ (green curves). This inspires us to start from an image prior with a lower number of latent channels and leave the remaining features for temporal compression, promoting a more balanced temporal-spatial compression initialization. To this end, we propose a dual-branch keyframe-based temporal compression architecture, where a 2D branch inheriting the low-dimensional image VAE prior focuses on keyframe compression, and a 3D branch faciliates temporal compression.

Second, the unidirectional information flow of 3D causal convolution prevents the interaction between video frames, exacerbating inter-frame unbalanced performance under temporal compression. For $M$ frames (a frame group) upsampled from the same compressed frame, the front frame cannot interact with other frames in the group due to the forward flow of causal convolution, while the last frame can acquire the information of the whole frame group, resulting in the preference of later frames in the optimization process. As shown in Fig. 1(b), the methods (Yang et al., 2024; hpcaitech, 2024; Chen et al., 2024) based on causal convolution suffer from significant inter-frame flicker, which is indicated by the fluctuating SSIM values across different frames. This inspires us to propose group causal convolution, which groups the input frames based on the temporal compression rate. Padding operations with causal logic are applied between groups to ensure the continuity in processing variable frame videos, and standard convolution operations are used within each group of $M$ frames to share equivalent interactive information, achieving smoother reconstruction results.

In addition, the local receptive field struggles to meet the same temporal compression demand at high resolutions. For the same video, the higher the resolution, the more pixels the same motion spans, making it more challenging for the model to capture motion patterns. To this end, we introduce dilated convolutions and added multiple attention modules to expand the receptive field, thus improving the temporal motion perception capability of the model at high resolutions.

Finally, considering the increasing attention to content diversity and resolution in the video generation field, there is a lack of a suitable open source benchmark for comprehensive evaluation of video VAE reconstruction capabilities. Most of the datasets (*e.g.*, Kinetics-600 (Carreira et al., 2018) and UCF-101 (Soomro et al., 2012)) are 720P or lower, and cannot be used for higher resolution (*e.g.*, 1080P) evaluations. In addition, high-resolution datasets usually consist mainly of slow-motion and fixed-shot videos, which hardly reflects the overall performance of video VAEs under varying motion speeds. Therefore, we re-collect a subset of 2000 videos with 1080P containing various motion speeds from the OpenVid-0.4M (Nan et al., 2024) dataset, named MotionHD, as a supplement to the overall evaluation. Experimental results show that our method achieves state-of-the-art video reconstruction at various resolutions and motion rates.

In summary, the contributions of this paper include: **(1)** We propose a keyframe-based temporal compression architecture to accelerate convergence speed by providing more balanced and powerful temporal-spatial compression initialization. **(2)** We introduce group causal convolution as a replacement for causal convolution to achieve better and balanced performance between frames, while maintaining causal logic. **(3)** Extensive experiments on five benchmark demonstrate the SOTA video reconstruction and generation capabilities of the proposed IV-VAE.

## 2 RELATED WORK

**Variational Autoencoder.** Variational Autoencoder (VAE), as a widely used model in the generation domain, adept at learning complex distributions from high-dimensional data. VAEs can be categorized into two types based on the class of tokens, including discrete and continuous. Classical work of the former type includes VQ-VAE (Van Den Oord et al., 2017), Magvit (Yu et al., 2023a) et al, where continuous latent features are discretized by quantization techniques for autoregression generation. The latter type of continuous VAE, which can achieve spatial or even temporal compression by mapping high-dimensional data distributions to a low-dimensional latent space, is widely used in latent diffusion models (Rombach et al., 2022; Blattmann et al., 2023).

**Video VAE.** Earlier video generation works (Zhou et al., 2024; Xu et al., 2024; Blattmann et al., 2023) usually use image VAE to compress video frames, or fine-tune the image VAE with additional time-dimensional convolutions to reduce flickering artifacts, but this approach cannot effectively reduce the redundancy in the time dimension. With the emergence of OpenAI's Sora (OpenAI, 2024), to allow video generation models to learn and generate longer videos, there have been some recent works (hpcaitech, 2024; Qin et al., 2024; Yang et al., 2024; Chen et al., 2024; Zhao et al., 2024) that attempt to train a video VAE with a larger time compression ratios (*e.g.*, 4x). Open-Sora (hpcaitech, 2024) adopts a stacked approach, first compressing the video frames in the spatial dimension based on SDXL image VAE (Podell et al., 2023), and then introducing 3D VAE to compress the temporal dimension, this practice dramatically reduces the GPU memory usage of video reconstruction. OD-VAE (Chen et al., 2024) uses tail initialization to inflate the 2D convolutions of SD image VAE (Rombach et al., 2022) into 3D causal convolutions and explores a more efficient model variant. CV-VAE (Zhao et al., 2024) uses latent space regularization to avoid latent space distribution shift while compressing time. CogVideoX (Yang et al., 2024) constructs a multi-GPU parallel computing approach to process long videos based on causal video VAE.

## 3 METHOD

In this section, we first construct a 3D causal VAE as the baseline method in Sec. 3.1. Then, we describe the proposed IV-VAE in detail, including group causal convolution (GCConv) in Sec. 3.2, keyframe-based temporal compression (KTC) in Sec. 3.3, and temporal motion perception enhancement (TMPE) in Sec. 3.4. The overall structure is shown in Fig. 2. Finally, we present a series of other improvements in the structure in Sec. 3.5.

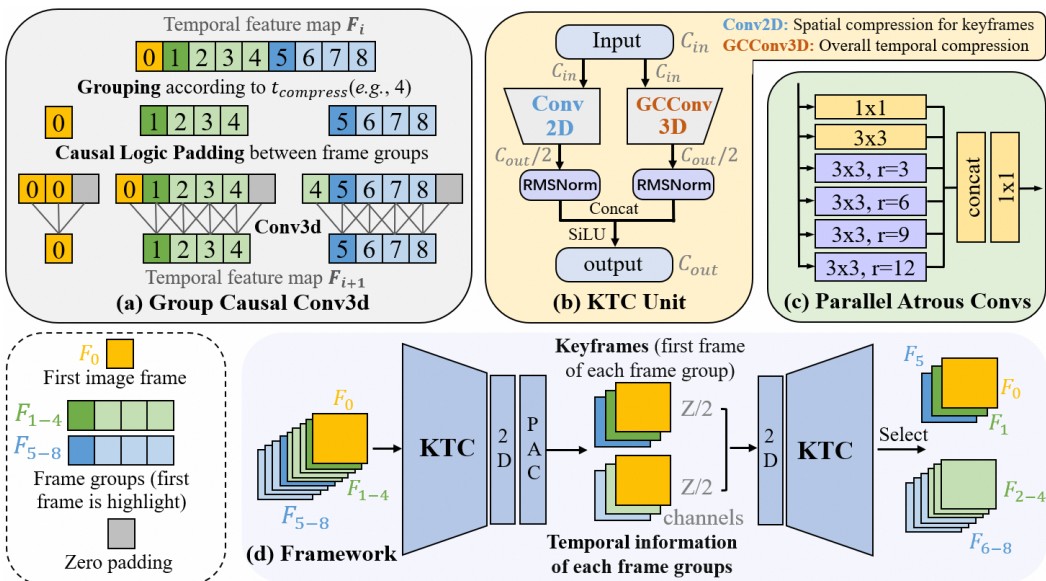

Figure 2: (a-c) **Basic components** of IV-VAE. (d) **Overall Framework**.

## 3.1 3D CAUSAL VAE

In the beginning, we introduce a causal video VAE as the baseline method. We first train an image VAE based on the architecture of the SD VAE with a few structural modifications (see subsection 3.5). Following the thought of Magvitv2 (Yu et al., 2023b), we choose to use 3D causal convolution in the construction of video VAE. Drawing on the idea of central initialization (Carreira & Zisserman, 2017), we inflate the 2D convolution in the image VAE to the 3D causal convolution. For example, noting the 2D convolutional weights as $W_{2D}$, when inflating $W_{2D}$ to be $W_{3D}$ with convolutional kernel time dimension of 3, $W_{3D}$ is formulated as $[0, 0, W_{2D}]$. In this way, we can obtain high image quality reconstruction results after extending the image VAE to a causal video VAE. Lastly, in addition to the first image frame being processed separately, the causal video VAE achieves 4x temporal compression on the input, following Magvitv2 (Yu et al., 2023b).

Adopting a causal VAE as the baseline approach has the following two advantages: (1) The first frame is always independent of other frames. According to Magvitv2 (Yu et al., 2023b), using a temporal causal VAE instead of a regular VAE makes the model perform better in processing a single image. (2) Each frame depends only on the previous frame. Thus, the temporal causal video VAE can use cache mechanism in the inference, which can split the reconstruction process of a long video into a series of shorter videos with exactly the same result. We consider this a better operation compared to the regular overleap approach, more details are displayed in subsection 4.5.2.

## 3.2 GROUP CAUSAL CONVOLUTION

Based on the above baseline, we note that it is practically impossible and non-optimal to ensure causal logic for all frames. For example, the baseline approach compresses time by a factor of 4, then the input 4 frames will be compressed into 1 frame by the encoder and then recovered into 4 frames through the decoder. Obviously, these 4 frames recovered from the same compressed frame do not satisfy the temporal causal correlation. In addition, for these 4 reconstructed frames, the performance of the first frame is usually worse than that of the following frames as shown in Fig. 1(b), resulting in frequent flickering in the reconstructed video. We attribute this problem to the insufficient and unbalanced interaction between neighboring frames due to the causal convolution.

Therefore, to alleviate this problem while being able to maintain the advantages of causal VAE (subsection 3.1), we propose group causal convolution (GCConv). As shown in Fig. 2(a), for a video VAE with a temporal compression rate of $t_c$ ($t_c = 4$ in our method), the proposed group causal convolution sequentially groups every $t_c$ frames of the input video frames into one frame group. In particular, the first image frame is a separate frame group. For a frame group, we aim to make the features between frames visible and interactable with each other, thus increasing the

reconstruction smoothness between frames and reducing the flicker. For different frame groups, we maintain the property of causal convolution that the current frame group only depends on the previous frame group. Therefore, the time padding scheme for group causal convolution is designed in Fig. 2(a). We pad the features of the previous frame before each frame group, in particular, the first image frame is padded using its own features. And we use zero padding after each frame group. After causal logic padding, each group is operated separately using the same regular convolution. Notably, the number of frames in the frame group is not fixed during the forward process, and for each temporal downsampling/upsampling operation of the feature map, the number of frames contained in the frame group decreases/increases by the same factor.

### 3.3 KEYFRAME-BASED TEMPORAL COMPRESSION

Building on the observation from Fig. 1(a), to obtain better temporal-spatial compression capability at the initialization of video VAE, it is feasible to utilize pre-trained weights of images with a low latent channel number. To this end, we propose a dual-branch keyframe-based temporal compression (KTC) architecture that decouples the compression of a frame group into the spatial compression of key frame and the temporal compression of the overall motion. The basic unit is shown in Fig. 2(b). For the input feature map $F$ with $C_{in}$ channels, it is fed into a 2D convolution to extract the image information and a 3D group causal convolution to extract the overall temporal motion information, respectively. The two convolutions each output feature maps with channel number $C_{out}/2$, which are then separately normalized by RMS-Norm and concatenated together. In the initialization phase, the 2D and 3D branches of the network are initialized using pre-trained image weights with the number of latent channels $Z/2$ respectively. In the 3d branch, the 2D image weights are inflated to 3d weights based on the center initialization (Carreira & Zisserman, 2017). In addition, by elaborately setting the weights of the newly added temporal upsampling and downsampling layers of the two branches, the network is initially able to independently process two different frames within a frame group to achieve 2× temporal compression with strong image compression capability as well.

The overall structure of the network is based on UNet (Ronneberger et al., 2015) and adopts the proposed KTC as the base unit, as shown in Fig. 2(d). After two temporal downsampling, a frame group is merged into a compressed frame, so we replace the GCConv3D in KTC Unit with a 2D convolution at this stage to reduce the computation. In the output of the model, each of the two branches produces a sequence of video frames with the same shape as the input, where the reconstruction results of keyframes and remaining frames are selected from the 2D branch and 3D branch outputs, respectively, and compose the final reconstruction result.

### 3.4 TEMPORAL MOTION PERCEPTION ENHANCEMENT

For the same video, as the video resolution increases, the pixel span generated by object motion or camera motion increases, and the more difficult it is for the video VAE to capture the motion patterns due to the limitations of the receptive field. To this end, we make two improvements to increase the receptive field of the video VAE to enhance the perception of temporal motion at high resolution. As shown in Fig. 2(c), we introduce multiple parallel atrous convolutions (PAC) with different atrous rates in the last layer of the encoder. The generated features of different branches are concatenated together and then pass through a $1 \times 1$ convolution to adjust the number of channels, borrowing from the ASPP in DeepLabV3+ (Chen et al., 2018). In addition, we expand the number of attention modules from 2 to 7, and all attention modules are implemented after full temporal and spatial compression to reduce the computational effort. At this time, the compressed frame contains all the information of the frame group, and the global receptive field of attention modules can effectively facilitate the perception of the temporal motion especially at high resolution.

### 3.5 OTHER STRUCTURAL MODIFICATIONS

Based on the architecture of the SD image VAE, we make some structural modifications to better adapt to the 3D causal VAE. First, we replace all GroupNorms with RMSNorms to ensure temporal causality following Magvitv2, since the standard GroupNorm involves temporal dimension when calculating the mean and variance of the features within each group, which destroys temporal causality. Second, we shrink the channel number during spatial upsampling instead of in the later residual block, which allows IV-VAE to reduce max memory usage by 29% in 480P video reconstruction.

## 4 EXPERIMENTS

### 4.1 MOTIONHD DATASET.

To better evaluate the reconstruction ability of video VAEs, we propose to consider the video reconstruction performance at different resolutions and different motion amplitudes. However, existing open-source datasets do not perfectly satisfy these two requirements, *e.g.*, the video resolution of the Kinetics-600, and UCF-101 datasets is not large enough (less than 1080P). The OpenVid-0.4M (Nan et al., 2024) dataset has a huge number of high quality videos at 1080p resolution, but most of them are fixed camera and slow moving. To better present this phenomenon, we use RAFT model (Teed & Deng, 2020) to compute the optical flow map of the video at 768x432 resolution and then get the average optical flow score as the motion score. As shown in Fig. 3, we present the motion distributions of Kinetics-600 and OpenVid-0.4M datasets, respectively. Compared to Kinetics-600, the motion distribution of OpenVid-0.4M is overall narrower and concentrated in the low motion interval, which is difficult to reflect the performance on videos with medium and fast motion. More importantly, in the training of diffusion models, it is common to use a high frame interval to facilitate the model to learn to the motion patterns, *e.g.* Latte set the frame interval to 3 on all datasets. In this case, the motion of the video will be more intense, as shown in Fig. 3(a).

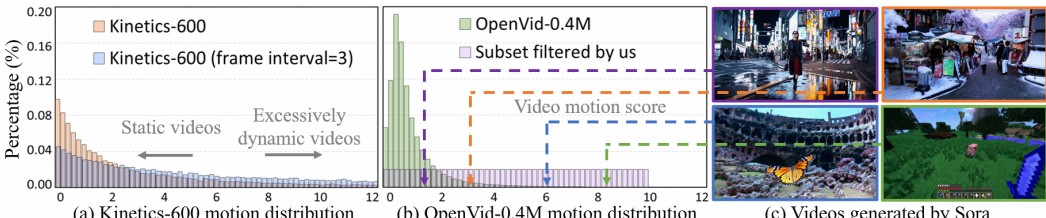

Figure 3: **Video motion distribution** of Kinetics-600 and OpenVid-0.4M.

Therefore, we refer to the motion distribution of the Kinetics-600 ( frame interval of 3), and uniformly select 2000 videos with 1080P resolution from OpenVid-0.4M according to the motion scores, as illustrated in Fig. 3(b), constituting the subset MotionHD, which is used to test the average performance of video VAEs at various video motion speeds and the performance at different resolutions. We also list the motion scores corresponding to the videos generated by OpenAI's SORA, to facilitate the interpretation of scores and to indicate that our choice of motion intervals is reasonable.

### 4.2 EXPERIMENTAL SETUP

**Evaluation details.** We evaluate the reconstruction performance of video VAEs on multiple video benchmarks, including Kinetics-600 (Carreira et al., 2018) validation set, ActivityNet (Caba Heilbron et al., 2015) test set, and MotionHD. For Kinetics-600 and ActivityNet, we each randomly select 2400 videos and test them with the original video resolution. For MotionHD, which is used to test the performance of Video VAEs at different resolutions, we first scale the video to the target resolution along the short side and then perform center cropping. All experiments are conducted using 17 frames of the video. To measure the reconstruction quality, structural similarity index measure (SSIM) (Wang et al., 2004), peak signal-to-noise ratio (PSNR), Learned Perceptual Image Patch Similarity (LPIPS) (Zhang et al., 2018), and Frechet Video Distance (FVD) (Unterthiner et al., 2018) are adopted as metrics.

Following OD-VAE (Chen et al., 2024), we employ Latte (Ma et al., 2024), a sora-like transformer-based diffusion model for video generation. We choose Kinetics-400 dataset (Kay et al., 2017) for class-conditional generation and the SkyTimelapse dataset (Xiong et al., 2018) for unconditional generation. All methods are trained for 150k steps with the same settings, where the parameters of the video are set to a resolution of 256×256 and a frame length of 17 during both training and testing. We employ FVD as the metric and calculate it on 2048 samples.

**Training details.** We first train an image VAE on 256×256 image data for 200k steps using 4 A800 GPUs, and then inflate the 2D convolutions to 3D convolutions based on center initialization (Carreira & Zisserman, 2017). Next we train the video VAE on 256×256 resolution videos for 500K steps, and further expand the training resolution to 512×512 training 200K steps. Finally,

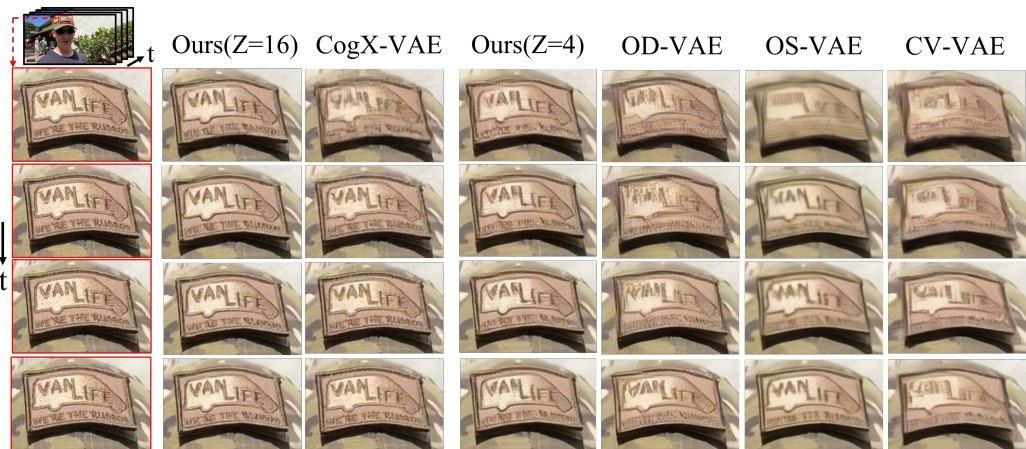

Figure 4: **Reconstruction results of different methods for a frame group.**

Table 1: **Video reconstruction results comparison**. Chn: Latent channel number. FCR: Frame compression rate.

| Method | Params | FCR | Chn | Kinetics-600 | | | | ActivityNet | | | |
|--------|--------|-----|-----|------|------|------|-------|------|------|------|-------|
| | | | | FVD↓ | PSNR↑ | SSIM↑ | LPIPS↓ | FVD↓ | PSNR↑ | SSIM↑ | LPIPS↓ |
| SVD VAE | 97M | 1*8*8 | 4 | 6.26 | 35.26 | 0.9363 | 0.03675 | 5.47 | 37.07 | 0.9298 | 0.03217 |
| CV-VAE | 182M | 4*8*8 | 4 | 16.81 | 31.58 | 0.9066 | 0.08271 | 12.57 | 34.87 | 0.9156 | 0.06846 |
| OS VAE | 393M | 4*8*8 | 4 | 19.05 | **34.37** | 0.9258 | 0.08162 | 13.56 | 37.64 | 0.9340 | 0.06781 |
| OD-VAE | 239M | 4*8*8 | 4 | 10.69 | 33.88 | 0.9242 | 0.05485 | 8.10 | 36.95 | 0.9321 | 0.04670 |
| Causal VAE | 127M | 4*8*8 | 4 | 10.81 | 33.77 | 0.9248 | 0.06309 | 8.26 | 37.06 | 0.9326 | 0.05336 |
| IV-VAE | **107M** | 4*8*8 | 4 | **8.01** | 34.29 | **0.9281** | **0.05209** | **6.08** | **38.47** | **0.9359** | **0.04436** |
| Causal VAE | 127M | 4*8*8 | 8 | 5.72 | 36.08 | 0.9479 | 0.04033 | 4.39 | 38.36 | 0.9519 | 0.03458 |
| IV-VAE | **107M** | 4*8*8 | 8 | **3.94** | **36.66** | **0.9520** | **0.03599** | **3.50** | **39.61** | **0.9566** | **0.03072** |
| CogX-VAE | 215M | 4*8*8 | 16 | 3.17 | 38.38 | 0.9677 | 0.02866 | 2.17 | 40.68 | 0.9703 | 0.02468 |
| Causal VAE | 127M | 4*8*8 | 16 | 3.28 | 38.07 | 0.9641 | 0.02887 | 2.43 | 40.42 | 0.9671 | 0.02483 |
| IV-VAE | **107M** | 4*8*8 | 16 | **2.97** | **39.02** | **0.9685** | **0.02280** | **2.01** | **42.61** | **0.9722** | **0.01968** |

we train the VAE at different resolutions with different frame numbers for 100K steps. GAN loss from 3D discriminator is added at this stage, and the training at different resolutions corresponds to different GAN discriminator weights. For the baseline causal VAE, we set the basic channel number of the convolution to 96. For our method, we set it to 64 for both 2D convolution and 3D GCConv.

## 4.3 PERFORMANCE

In addition to our baseline method Causal VAE, comparison methods include OpenSora V1.2 (OS) VAE (hpcaitech, 2024), OD-VAE V1.2 (Chen et al., 2024), CV-VAE (Zhao et al., 2024), CogVideoX (CogX) VAE (Yang et al., 2024), SVD VAE (Blattmann et al., 2023).

**Quantitative evaluation of reconstruction results.** Table 1 demonstrates the performance of the proposed Causal VAE and IV-VAE with other methods on Kinetics-600 and ActivityNet datasets. Experiments show that the proposed method achieves excellent results on both datasets and various numbers of latent channels $Z$. On $Z = 4$, we outperform OD-VAE by 2.68 and 2.02 FVD in Kinetics-600 and ActivityNet, respectively and with less than half the number of parameters. Compared to OS-VAE, we are slightly lower on the PSNR metric due to the limitation of the number of parameters, but we achieve a reduction of 286M (-73%) parameters, 11.04 (-58%) FVD, and 0.02953 (-36%) LPIPS on Kinetics-600. Compared to SVD VAE, although our model with $Z = 8$ use a latent channel number twice as large, we achieve an additional 4x time compression and perform better on all metrics. In addition, we also obtain the best results on a latent channel number of 16, especially in the PSNR and LPIPS metrics.

**Qualitative evaluation of reconstruction results.** As shown in Fig. 4 and 5, we exhibit the reconstruction results within a frame group for the different methods. It can be observed that other

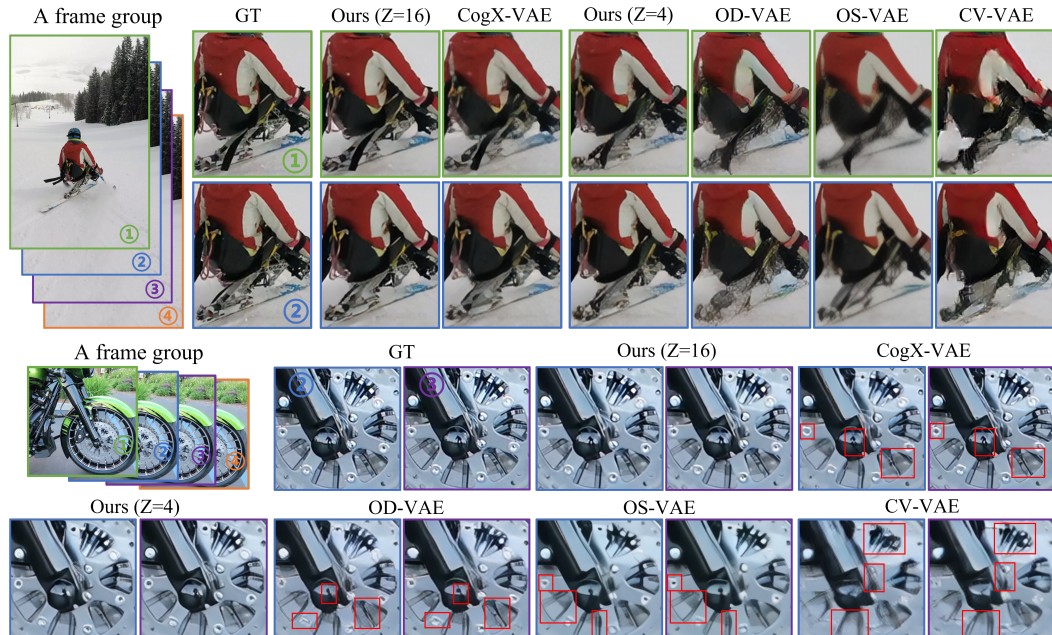

Figure 5: **Reconstruction results for two adjacent frames in a frame group.** In the below figure, the regions with large differences between the two reconstructed frames are boxed out.

Table 2: **Video reconstruction performance at different resolutions** on MotionHD dataset. **480P**: 848×480 resolution, **720P**: 1280×720 resolution, **1080P**: 1920×1280 resolution.

| Method | Chn | FVD↓ | | | PSNR↑ | | | SSIM↑ | | | LPIPS↓ | | |
|---|---|---|---|---|---|---|---|---|---|---|---|---|---|
| | | 480P | 720P | 1080P | 480P | 720P | 1080P | 480P | 720P | 1080P | 480P | 720P | 1080P |
| SVD VAE | 4 | 5.68 | 4.23 | 3.62 | 33.71 | 36.00 | 37.69 | 0.9313 | 0.9507 | 0.9617 | 0.035 | 0.033 | 0.031 |
| CV-VAE | 4 | 22.33 | 12.78 | 8.50 | 29.70 | 30.29 | 31.19 | 0.8796 | 0.9008 | 0.9206 | 0.094 | 0.098 | 0.098 |
| OS VAE | 4 | 16.89 | 10.06 | 8.21 | **32.33** | **34.13** | 35.65 | 0.9109 | 0.9310 | 0.9452 | 0.086 | 0.086 | 0.087 |
| OD-VAE | 4 | 10.40 | 7.36 | 5.31 | 31.42 | 33.32 | 34.89 | 0.9004 | 0.9226 | 0.9391 | 0.059 | 0.061 | 0.064 |
| Causal VAE | 4 | 9.33 | 6.85 | 5.18 | 31.64 | 33.43 | 34.82 | 0.9087 | 0.9286 | 0.9415 | 0.067 | 0.069 | 0.073 |
| IV-VAE | 4 | **7.01** | **4.49** | **4.31** | 32.01 | 34.08 | **35.80** | **0.9127** | **0.9333** | **0.9465** | **0.053** | **0.054** | **0.055** |
| Causal VAE | 8 | 4.63 | 4.15 | 3.58 | 34.14 | 35.89 | 37.15 | 0.9355 | 0.9481 | 0.9557 | 0.043 | 0.047 | 0.051 |
| IV-VAE | 8 | **4.09** | **3.71** | **2.86** | **34.60** | **36.56** | **38.11** | **0.9410** | **0.9534** | **0.9611** | **0.037** | **0.039** | **0.041** |
| CogX-VAE | 16 | 3.50 | 3.66 | 2.26 | 36.52 | 37.86 | 38.68 | 0.9571 | 0.9639 | 0.9676 | 0.031 | 0.035 | 0.039 |
| Causal VAE | 16 | 3.76 | 3.85 | 2.72 | 36.25 | 37.67 | 38.60 | 0.9544 | 0.9617 | 0.9663 | 0.032 | 0.037 | 0.041 |
| IV-VAE | 16 | **2.58** | **2.78** | **2.18** | **36.80** | **38.81** | **40.33** | **0.9585** | **0.9669** | **0.9720** | **0.025** | **0.027** | **0.027** |

methods usually perform poorly in the first two frames of a frame group, with blurring and a small amount of distortion, and the last frame performs better. This phenomenon validates our point that for a video VAE with temporal compression, the use of causal convolution makes the frames in the same frame group unequal with respect to the interaction of information, leading to performance imbalance. While our method can achieve a more balanced performance benefiting from group causal convolution. Besides, due to the performance imbalance, other methods usually suffer from abrupt changes in details in the reconstruction of neighboring frames, as shown in the lower panel of Fig. 5, and our method performs better in terms of temporal consistency and reconstruction quality.

**Reconstruction performance at different resolutions.** In Table 2, we present the performance of various methods at different resolutions using MotionHD dataset with uniform motion distribution. The proposed method achieves the best results in the majority of metrics. And as the resolution increases, our improvement relative to other methods is more pronounced. For example, in the PSNR metric, we outperform CogX-VAE by 0.28 at 480P while when the resolution is 1080P, this improvement increases to 1.65. Experiments indicate that the proposed KTC module and TMPE can enable the video VAE to achieve better temporal compression capability, leading to better results in challenging scenarios of temporal compression such as high resolution and rapid motion.

Table 3: Left: **FVD for different video generation methods.** All methods use the same latent channel number $Z = 4$. Right: **Video frames generated by our method.** Sky: SkyTimelapse.

| Method | K400 | Sky |
|--------|------|-----|
| CV-VAE | 552.4 | 128.0 |
| OS-VAE | 617.6 | 141.3 |
| OD-VAE | 548.7 | 140.0 |
| **IV-VAE** | **541.6** | **118.6** |

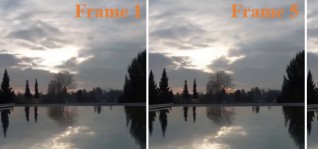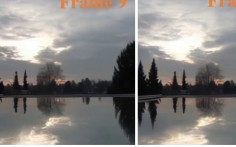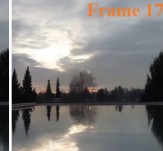

**Video Generation.** On the left side of table 3, we demonstrate the video generation performance of different video VAEs using the same generation model Latte (Ma et al., 2024). The proposed VAE achieves the best performance on both class-conditional and unconditional generation. Especially on the SkyTimelapse dataset, we obtain a 9.4 FVD decrease compared to CV-VAE. In the right figure of Table 3, we show some frames of the video generated by Latte with our VAE ($Z = 4$). The frames exhibit a fairly high level of quality and realism, and even reflect some of the laws of physics, as the sunlight in the sky moves, so does the light reflected in the water.

Table 4: Left: **Ablation study**. (A) represents the baseline method Causal VAE. Right: **Training loss curves for different structures.** The training loss is the sum of MAE loss and LPIPS loss.

| Setting | | | Kinetics-600 val | | |
|---------|---|---|------------------|---|---|
| GCConv | KTC | TMPE | PSNR↑ | SSIM↑ | LPIPS↓ |
| **(A)** | | | 31.29 | 0.9042 | 0.05233 |
| **(B)** | ✓ | | 31.64 | 0.9082 | 0.05028 |
| **(C)** | | ✓ | 31.86 | 0.9116 | 0.04865 |
| **(D)** | ✓ | ✓ | 32.12 | 0.9145 | 0.04744 |
| **(E)** | ✓ | ✓ | ✓ | 32.24 | **0.9158** | **0.04705** |

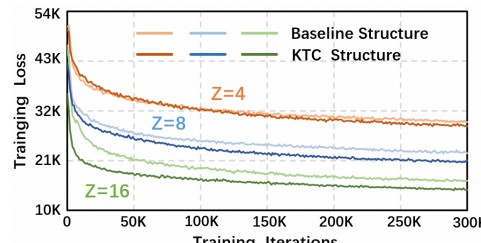

### 4.4 ABLATION STUDY

In this section, we perform a series of ablation experiments. Considering that ablation for three channel numbers ($Z = 4, 8, 16$) requires too much time and resources, we choose the middle value of the latent channel number $Z = 8$, for the main ablation experiment. For each ablation experiment we follow the same training procedure: we first train an image VAE for 200k steps, and then extend it to a video VAE and train it for 300K steps on 17-frame videos with $256 \times 256$ resolution. The test dataset uses 2400 videos from Kinetics-600 validation set with 17-frame at $256 \times 256$ resolution.

**Ablation studies of main components.** Based on the baseline causal VAE, we explore the effects of the different compositions of IV-VAE in Table 4. As can be seen from the table, the implementation of the proposed group causal convolution can substantially improve the performance on both baseline and KTC structure and do not increase the number of parameters. The results indicate that the reconstruction quality can be effectively improved by allowing the frames in the same frame group to interact bidirectionally. Further, replacing the base unit with KTC unit can substantially facilitate the learning of temporal compression thus improving the performance of the model. Finally, we add TMPE to enhance the model's temporal motion perception capability by increasing the global receptive field, which allows the model to better learn the object's motion patterns, with only 3M additional parameter counts (from 104M to 107M).

**Effectiveness of KTC at different numbers of latent channels.** To verify the validity of the KTC structure, we only replace the structure of the baseline causal VAE with the proposed KTC and perform experiments using the same training setup, the training loss is shown in the right panel of Table 4. It can be seen that when the number of latent channels $Z = 4$, the proposed KTC performs slightly superior to the baseline as the training steps increase. And this advantage extends significantly when $Z$ expands to 8 and 16. In conjunction with the results presented in Fig. 1, it can be analyzed that the spatial compression improvement of the image VAE is attenuated as $Z$ increases. Therefore, when $Z$ is large, dedicating a portion of the number of latent channels to the initial temporal compression can achieve better and balanced temporal-spatial compression capability initially, leading to faster convergence in subsequent training.

**Why choose a dual branch of 2D and 3D.** We ablate the branch structure of the KTC unit by replacing the conv2D with GCConv3D. The experimental results are listed in Table 5, compared to the original 2D+3D structure, using the 3D+3D structure can achieve a slight enhancements. However, it also increases the parameter count by 23%, so we finally chose 2D+3D as a more efficient structure.

Table 5: **Ablation for branch selection.**

|  | Params | PSNR↑ | SSIM↑ | LPIPS↓ |
|---|---|---|---|---|
| (2D+3D) | **107M** | 32.24 | 0.9158 | 0.04725 |
| (3D+3D) | 132M | **32.31** | **0.9163** | **0.04704** |

### 4.5 ANALYSIS

#### 4.5.1 INCREASE THE NUMBER OF PARAMETERS.

In Table 6, we explore the enhancements provided by increasing the parameters. We increase the base feature dimension from 64 to 96, which dramatically increases the parameters and also brings significant improvements especially in PSNR and LPIPS metrics. Compared to CogX-VAE, this model achieves a 1.19 PSNR improvement and still takes less time to reconstruct the video.

Table 6: **Effect of increasing the number of parameters**.

| Method | Params | Kinetics-600 | | | A 25-frame 1024×1024 Video Reconstruction Time |
|---|---|---|---|---|---|
| | | PSNR↑ | SSIM↑ | LPIPS↓ | |
| CogX-VAE | 215M | 38.38 | 0.9677 | 0.02866 | 11.4s |
| Ours (Z=16, dim=64) | 107M | 39.02 | 0.9685 | 0.02280 | **6.2s** |
| Ours (Z=16, dim=96) | 242M | **39.57** | **0.9711** | **0.02054** | 10.7s |

#### 4.5.2 CACHE MECHANISM $vs$ OVERLEAP MECHANISM.

For the reconstruction of a long video, it is usually impractical to encode and decode it in a single operation due to the limitation of the memory, therefore it is necessary to reconstruct the original video by splitting it into a number of clips. Overlap is the most commonly used approach (Chen et al., 2024), where each input video clip has a one-frame overlap with the previous clip as a history frame, and the overlapped frame is discarded after reconstruction. However, encoding through overlap is not equivalent to single-step encoding, which may introduce semantic bias in the latent space due to short history frame. And the existence of overlapped frame leads to extra computation.

To address this problem, we introduce the cache mechanism, which is based on the property that the causal network depends only on the previous frames in temporal order. According to the length $P_t$ needed for 3D convolutional temporal padding, the last $P_t$ frames of the current video clip's feature maps are saved at each layer of the network, which are used for the convolutional padding of the next video clip. Therefore, this mechanism can be fully equivalent to one-step reconstruction in terms of results. In addition, as shown in Table 7, the use of the caching mechanism achieves less time usage and video memory usage compared to the overlap mechanism.

Table 7: **Cache mechanism $vs$ Overleap mechanism.**

| Reconstruction approach | Input: A 49-frame 1024×1024 Video | | |
|---|---|---|---|
| | Video clipping | Memory Usage | Time Usage |
| Single-Step | 49-frame * 1 | 76.3G | **12.7s** |
| Overleap mechanism | 5-frame * 12 | 14.3G | 14.7s |
| Cache mechanism | 1-frame + 4-frame*12 | **14.1G** | 13.0s |

## 5 LIMITATION.

The overall architecture of the proposed method is still based on UNet following SD image VAE without exploring on other architectures. Video VAE faces more and unique difficulties compared to image VAE, *e.g.*, as the video resolution increases, the need for receptive field increases for video VAE. While the classical UNet lacks a global receptive field, in addition, the number of spatial downsampling of video VAE is usually aligned with the spatial compression ratio, which also limits the receptive field of video VAE. Therefore, it is worthwhile to consider introducing new architectures such as Dit or Mamba into video VAE in future work.

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

# A APPENDIX

## A.1 CALCULATION OF THE INFORMATION PRESERVATION DEGREE

To measure the information completeness of the video after compression using different VAE models, we borrow from the way of calculating the information loss rate of the dimensionality data in Principal Component Analysis (PCA), which is formulated as:

$$\frac{\sum_i^m \|x_i - x'_i\|^2}{\sum_i^m \|x_i\|^2}, \tag{1}$$

where $x_i$ is the original video, $x'_i$ is the reconstructed video and $m$ is the number of samples. However, considering that MAE loss is used in the training of VAE model, we adopt the following formula to measure the information preservation degree:

$$1 - \frac{\sum_i^m |x_i - x'_i|}{\sum_i^m |x_i|}. \tag{2}$$

## A.2 CALCULATION PROCESS OF SSIM RESULTS OF DIFFERENT FRAMES WITHIN A FRAME GROUP

To measure the performance of frames with different positions in each frame group, we calculate the SSIM results for different frames in the 17 reconstructed frames on the Kinetics-600 validation set. After excluding the first image frame, the remaining 16 frames contain 4 frame groups. We calculate the average performance of frames at the same position in different frame groups, *e.g.*, the first frame of each frame group, whose positions in the 16 frames are 1, 5, 9, and 13, respectively. In this way, we obtain the average performance of each position in a group of frames for a not-so-short time series, to ensure the generalization of the experimental results.

## A.3 OTHER RECONSTRUCTION RESULTS

In Fig. 6, we show the reconstruction results of the different methods within a frame group. It can be seen that we have a more balanced and stable performance over four consecutive frames, whereas the other methods have a very large difference in performance between frames, resulting in noticeable flickering.

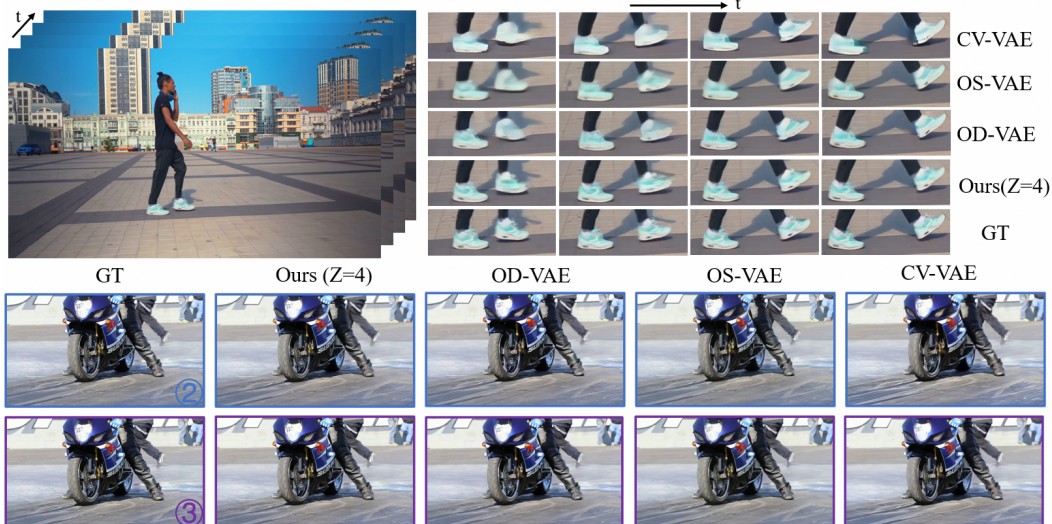

Figure 6: **Comparison of reconstruction results.**

