# OpenReview forum: "Improved Video VAE for Latent Video Diffusion Model"
_ICLR.cc/2025/Conference — ICLR 2025 Conference Withdrawn Submission_

### Official Review · Reviewer_RwZv · 2024-11-03

**Soundness:** 3
**Presentation:** 2
**Contribution:** 3
**Rating:** 6
**Confidence:** 4

**Summary:**

This paper presents several architecture modifications for video VAE typically used for latent video diffusion model training. The modifications are based on two observations: (a) a naive initialization from image VAE that uses the same latent dimension for video VAE is suboptimal, (b) and the per-frame reconstruction within the same block processed per each causal step (e.g., 4 frames) is not equal. Thus, the authors introduce an alternative initialization that uses half of the latent dimension for spatial compression and half of the latent dimension for temporal compression. They also propose several blocks to mitigate different per-frame compression in the same block of consecutive frames. Based on these modifications, the authors argue the proposed architecture outperform previous video VAEs and can improve the video generation performance.

**Strengths:**

- The paper is well-motivated, and the observations are interesting (although these might be quite trivial).
- The proposed method seems to show clear improvement compared with existing video VAEs.
- A new benchmark to evaluate video VAEs on high-resolution and dynamic video is interesting.
- The paper tries to demonstrate the effectiveness of the proposed VAE in video generation, not limited to showing better video reconstruction.

**Weaknesses:**

- Overall, I am positive about the paper, but in its current status, I cannot give it a very high rating because my contribution to the community is a bit marginal. Specifically, although the paper shows an improved performance compared with baselines under the same compression ratio, the proposed method does not compress the video more extensively (like more temporal compression than 4x). It can improve the video generation quality slightly (as shown by video generation results on Kinetics-600 and Skytimelapse), but the proposed video VAE still suffers from a fundamental problem of video generation models: handling memory or computation burdens. That being said, I think some observations and architectural modifications will be interesting to the community.
- Due to the increased number of layers (e.g., 2D + 3D in KTC unit), I expect the encoding/decoding time would be slower than other autoencoder variants. The paper seems to compare the time only with CogX-VAE on Table 6, but I think this part should be analyzed more extensively.
- The hyperparameter details for the diffusion model (using Latte) experiments are missing
- Visualizing videos as a stack of image frames is really a bad direction to visualize the video results; could the authors provide exact video files in the supplementary material or anonymized site? I've observed so many cases that two of them are not corresponds very well to each other.
- Also qualitative comparison on video reconstruction and generation is insufficient; could the authors add more examples?

**Questions:**

- Will the authors release the VAE checkpoint and code for reproducibility?

---

### Official Review · Reviewer_NecU · 2024-11-04

**Soundness:** 3
**Presentation:** 3
**Contribution:** 2
**Rating:** 5
**Confidence:** 4

**Summary:**

Previous video VAE methods mainly freeze the weights of 2D image VAE and then train the temporal decoder covering 3D convolutional blocks to achieve inter-frame feature interaction. However, there is still the challenge of temporal dimension redundancy. This work proposes a method to improve video VAE through structural design and algorithm optimization, aiming to solve the problems of insufficient time compression and frame imbalance in existing video generation models.

**Strengths:**

1.	This work proposes a new keyframe-based temporal compression method, which can make the video VAE converge faster and reconstruct better by providing a better initialization.
2.	The proposed grouped causal convolution method to solve the problem of unbalanced interaction of features in the temporal dimension and unstable performance caused by the previous reliance on causal convolution.
3.	To solve the problem of insufficient receptive field of VAE when compressing and reconstructing high-resolution videos, this work added a new PAC module to VAE and increased the number of attention modules.
4.	The proposed a cache-based mechanism enables VAE to better process long-duration videos in segments.

**Weaknesses:**

1.  The main purpose of video VAE is to serve the video diffusion model. In addition to training Latte with different VAEs on two video datasets and then evaluating the FVD of videos generated by models trained by different VAEs, this article has no other test indicators related to video generation to prove that the proposed IV-VAE can improve the generation effect of the video diffusion model or reduce the hardware resources required for training or reasoning of the video diffusion model. At the same time, the author completely lacks quantitative evaluation of the impact of VAE on video generation quality and video expansion model training and reasoning in the ablation experiment.
2.  According to the results of the author's ablation experiment in Table 4, the proposed TMPE has limited improvement in video reconstruction quality, but introduces additional computing power overhead. It is impossible to prove whether this slight improvement in reconstruction performance is due to the effectiveness of the TMPE algorithm itself or simply due to increased computing power.
3.  The cache-based mechanism proposed by the author enables video VAE to process long videos with lower hardware overhead, but there is a lack of quantitative comparison of video reconstruction and generation quality.

**Questions:**

The main shortcoming of this article is that the experiments are not enough to fully demonstrate the effectiveness of the proposed method.

---

### Official Review · Reviewer_mLKe · 2024-11-04

**Soundness:** 3
**Presentation:** 2
**Contribution:** 2
**Rating:** 5
**Confidence:** 5

**Summary:**

This paper presents an improved video VAE (IV-VAE) model designed to enhance spatio-temporal compression and reconstruction quality in latent video diffusion models. It proposes a Keyframe-based Temporal Compression (KTC) architecture and a Group Causal Convolution (GCConv) mechanism. These components address limitations of conventional 3D VAEs by improving temporal compression initialization, reducing inter-frame inconsistency, and increasing the receptive field to better capture motion across high-resolution frames. Extensive evaluations on datasets such as Kinetics-600 and MotionHD demonstrate IV-VAE’s improved spatio-temporal consistency, frame fidelity, and resistance to temporal flickering compared to other state-of-the-art methods.

**Strengths:**

* The paper’s key contributions, namely the KTC and GCConv modules, demonstrate some improvements over conventional approaches in both compression quality and temporal consistency. By dedicating latent channels for keyframes, IV-VAE provides a balanced spatiotemporal initialization that improves compression, especially when compared to standard 3D-causal VAEs that struggle with frame flickering due to forward-only information flow.
* The introduction of the MotionHD dataset is a valuable contribution for future works, as it fills a gap in testing high-resolution and high-motion content, providing an essential benchmark for video VAE reconstruction capabilities.
* Quantitative comparisons, such as FVD, PSNR, and SSIM, highlight IV-VAE’s effectiveness across various datasets, showing that it surpasses models like OD-VAE and OS-VAE with significantly fewer parameters.

**Weaknesses:**

* Missing Comparisons with Recent Works: The paper lacks citation and comparative discussion of related work like Video Probabilistic Diffusion Models (Video Probabilistic Diffusion Models in Projected Latent Space CVPR 2024) and Hybrid Video Diffusion Models (Hybrid Video Diffusion Models with 2D Triplane and 3D Wavelet Representation, ECCV 2024). Including references to these models and their qualitative/quantitative results would strengthen the argument, especially since both focus on enhanced autoencoder backbones for video generation.

* Figure 1 Clarity Issues: Figure 1 is particularly hard to interpret due to insufficient labeling and unclear meaning of the dashed lines in various colors. For example, the significance of the green dashed arrow is not explicitly explained, which may lead to confusion regarding the model’s initialization and compression gains. Clarifying these lines would improve readers’ understanding of the proposed approach.

* Ambiguous Explanation in Introduction: The fifth paragraph of the Introduction is challenging to follow, and Figure 1(b) does not align with the accompanying text. For instance, it is difficult to reconcile why CogX-VAE’s third frame outperforms other frames or why OS-VAE’s second frame performs better than its third. This discrepancy suggests a need for clarification or additional discussion regarding these frame-level performance inconsistencies.

* Inconsistent Model References: The model names, such as OD-VAE, OS-VAE, and CogX-VAE, are not clearly linked to their respective citations until later sections, like 4.3. This misalignment disrupts the reader’s ability to match models with their attributes or methods and should be addressed earlier for clarity.

* Technical Ambiguities in Section 3.1: The notation [0, 0, W_{2D}] for W_{3D} is unclear, particularly as it relates to causal padding and convolution. The authors could improve this by specifying whether these are zero matrices or zero-value vectors, which would make the method implementation more understandable.

* Metric Reporting and Alignment with Core Objective: Given the focus on spatiotemporal quality, metrics specifically targeting spatio-temporal fidelity must be also compared, such as STREAM (ICLR 2024 STREAM: Spatio-TempoRal Evaluation and Analysis Metric for Video Generative Models). As this metric evaluates both spatial and temporal coherence, its inclusion would further validate the model’s strengths in these dimensions.

* Ambiguity in Frame Group Description (Section 3.2): What does the statement, "The number of frames in the frame group is not fixed during the forward pass," mean? Doesn't the t_c = 4 setup mean a fixed number (4) of frames in the frame group?.

* Please mention the fact that the first frame acts as a keyframe in the text (Section 3.3). Currently, I believe it is mentioned only in a figure. Additionally, is Z/2 = C_{\text{out}}? Also, please clarify the exact mechanism for assigning weights to temporal upsampling and downsampling layers so that readers can understand why and how it enables the network to independently process two different frames within a frame group.

* In Section 3.3, why is it needed to maintain two branches after converting GCConv3D to 2D after two temporal downsampling stages?

* Please consider using other examples and visualization for the qualitative comparison to show the differences better. Currently, it is hard to notice the differences.

* Please use the original name format for each model. For example, "Magvit" -> " MagViT".

* Will MotionHD dataset be available for public use? Please make sure it is available for reproducibility.

**Questions:**

Please see my comments in Weaknesses

---

### Note · Authors · 2024-11-14

I have read and agree with the venue's withdrawal policy on behalf of myself and my co-authors.